# A Variational Perspective on Generative Flow Networks

**Heiko Zimmermann**  *h.zimmermann@uva.nl*
*Amsterdam Machine Learning Lab,*
*University of Amsterdam*

**Fredrik Lindsten**  *fredrik.lindsten@liu.se*
*Division of Statistics and Machine Learning,*
*Linköping University*

**Jan-Willem van de Meent**  *j.w.vandemeent@uva.nl*
*Amsterdam Machine Learning Lab,*
*University of Amsterdam*

**Christian A. Naesseth**  *c.a.naesseth@uva.nl*
*Amsterdam Machine Learning Lab,*
*University of Amsterdam*

**Reviewed on OpenReview:** *https://openreview.net/forum?id=AZ4GobeSLq*

## Abstract

Generative flow networks (GFNs) are a class of probabilistic models for sequential sampling of composite objects, proportional to a target distribution that is defined in terms of an energy function or a reward. GFNs are typically trained using a flow matching or trajectory balance objective, which matches forward and backward transition models over trajectories. In this work we introduce a variational objective for training GFNs, which is a convex combination of the reverse- and forward KL divergences, and compare it to the trajectory balance objective when sampling from the forward- and backward model, respectively. We show that, in certain settings, variational inference for GFNs is equivalent to minimizing the trajectory balance objective, in the sense that both methods compute the same score-function gradient. This insight suggests that in these settings, control variates, which are commonly used to reduce the variance of score-function gradient estimates, can also be used with the trajectory balance objective. We evaluate our findings and the performance of the proposed variational objective numerically by comparing it to the trajectory balance objective on two synthetic tasks.

## 1 Introduction

Generative flow networks (GFNs) (Bengio et al., 2021a;b) have recently been proposed as a computationally efficient method for sampling composite objects such as molecule strings (Bengio et al., 2021a), DNA sequences (Jain et al., 2022) or graphs (Deleu et al., 2022). To generate such objects, GFNs sample a trajectory along a directed acyclic graph (DAG) in which edges correspond to actions that modify the object. A trajectory sequentially constructs an object by transitioning from a root node (initial object, or null state) to a terminating node (final composite object) which is scored according to a reward signal. While sampling sequences of actions has been well studied in the reinforcement learning literature (Sutton and Barto, 2018), the objective is typically to find a policy which maximizes the expected reward of the trajectory. By contrast, GFNs are trained to learn a policy that solves a planning-as-inference problem (Toussaint et al., 2006) by learning a distribution over trajectories ending in a terminating state with probability proportional to the reward assigned to it. This is done by optimizing objectives which aim to satisfy a flow matching or detailed

balance condition (Bengio et al., 2021a). Malkin et al. (2022b) has since found that these objectives are prone to ineffective credit propagation across trajectories and proposes an alternative objective based on a trajectory balance (TB) condition to alleviate these problems. Most recently, Madan et al. (2022) proposed an objective that can be optimized on partial trajectories and (Do et al., 2022) proposed an optimal-transport-based objective to further improve generalization and exploration.

A positive reward function can be interpreted as an unnormalized distribution, which one wishes to generate samples from. In this view, we are interested in sequentially sampling form a factorized joint distibution, such that the marginal distribution of the final state is approximately equal to the corresponding normalized distribution. Generating approximate samples from an unnormalized target distribution is a well-studied task in probabilistic inference, for which many methods have been developed. Examples include methods based on MCMC (Hoffman and Gelman, 2014; Salimans et al., 2015; Li et al., 2017; Hoffman, 2017; Naesseth et al., 2020; Zhang et al., 2022c), importance sampling Neal (2001); Del Moral et al. (2006); Naesseth et al. (2019) and variational inference (Blei et al., 2017; Naesseth et al., 2018; Maddison et al., 2017; Le et al., 2018; Zimmermann et al., 2021). Recent work on GFNs by Zhang et al. (2022b) takes a similar view by treating the reward function as an energy-based model, which can be trained to maximize the data likelihood following a contrastive divergence-based approach (Hinton, 2002), while the forward- and backward transition models are trained by optimizing the TB objective.

In this work we show that, in certain settings, optimizing the TB objective is equivalent to optimizing a forward- or reverse Kullback–Leibler divergence. To this end we compare the TB objective when optimized with samples generated from the forward transition model, backward transition model, or a mixture of both, to training with a corresponding variational objective, which takes the form of a convex combination of the forward- and reverse Kullback-Leibler divergence. We identify cases in which the TB objective is equivalent to the corresponding variational objective and leverage this insight to employ variance reduction techniques from variational inference. Finally, we run experiments, to evaluate our theoretical findings and the empirical performance of the trajectory balance and the corresponding variational objective.

**Related Work** Recent work by Zhang et al. (2022a) identifies equivalences between GFNs and certain classes of generative models. The authors observe that hierarchical variational auto-encoders are equivalent to a special class of GFNs, and that training hierarchical latent variable models with the forward KL divergence between the full backward- and forward transition model of the GFN is equivalent to training a hierarchical VAE by maximizing its ELBO.

In concurrent and independent work, Malkin et al. (2022b) derive the same equivalences between optimizing the TB objective and forward- and reverse KL divergence that we establish in this work. The difference with our work is that we establish these equivalences based on a composite objective which is a convex combination of the reverse and forward Kullback-Leibler divergences. Using the composite objective we also explore setting for which training is not equivalent but conceptually similar in the sense that the same proportion of samples is taken from the forward and backward model. Furthermore, we discuss and study this objective in context of learning energy-based models. Finally, we also study the differences between variational inference and trajectory balance optimization when the forward and backward trajectory distributions share parameters.

## 2 Background

In the following we give a brief introduction to Generative Flow Networks, variational inference, and variance reduction techniques for discrete variable models. It is important to note that we introduce the concept of flows, which are used to define the a forward- and reverse transition model based on the *flow* along the edges of a DAG, for completeness only. In the remainder we only assume the existence of a proper forward- and backward transition model for a given DAG, which does not need to be parameterized by a flow. For a comprehensive study of flows and generative flow networks we refer to Bengio et al. (2021a;b).

### 2.1 Generative Flow Networks

Generative Flow Networks (Bengio et al., 2021a) aim to generate trajectories $\tau = (s_0, s_1, \ldots, s_T, s_f)$ along the edges of a directed acyclic graph $G = (\mathcal{S}, E)$. Each trajectory starts in the root, $s_0$, and terminates in a *terminating state*, $s_T$, before transitioning to a special *final state*, $s_f$, which is the single leaf node of $G$. A non-negative reward signal $R(s_T)$ is assigned to each terminating state $s_T$. The task is to learn a sampling procedure, or flow, for simulating trajectories, such that the marginal distribution of reaching the terminating state $s_T$ is proportional to $R(s_T)$. We adopt the convention that $s_f = s_{T+1}$. The structure of the DAG imposes a partial order, $<$, on states $s, s' \in \mathcal{S}$ such that $s < s'$ if $s$ is an ancestor of $s'$. Hence, any trajectory satisfies $s_j < s_k$ for $0 \leq j < k \leq T+1$ and consequently does not contain loops. In Figure 1 we illustrate a possible DAG structure and corresponding forward- and backward transition models over the domain of (extended) binary vectors.

#### 2.1.1 Trajectory Flows

A *trajectory flow* is a non-negative function $F_G : \mathcal{T} \to \mathbb{R}^+$ on complete trajectories $\mathcal{T}$, i.e. trajectories starting in a *initial state* $s_0$ and ending in the final state $s_f$ associated with a DAG $G$. Below, we drop the graph subscript for notational convenience. A trajectory flow defines a probability measure $P$ over complete trajectories, such that for any event $A \subseteq \mathcal{T}$

$$P(A) = \frac{F(A)}{Z}, \qquad F(A) = \sum_{\tau \in A} F(\tau), \qquad Z = \sum_{\tau \in \mathcal{T}} F(\tau),$$

where $Z$ can be interpreted as the total amount of flow. The flow $F(s)$ through a state and the flow $F(s \to s')$ along an edge $(s, s')$ are denoted by

$$F(s) := F(\{\tau \in \mathcal{T} : s \in \tau\}), \qquad F(s \to s') := F(\{\tau \in \mathcal{T} \mid \exists t \in \mathbb{N} : s = s_t, s' = s_{t+1} \in \tau\}).$$

The probability of a trajectory containing the state $s$, and the *forward-* and *backward transition probabilities* are denoted by

$$P(s) := \frac{F(s)}{Z}, \quad P_F(s' \mid s) := P(s \to s' \mid s) := \frac{F(s \to s')}{F(s)}, \quad P_B(s \mid s') := P(s \to s' \mid s') = \frac{F(s \to s')}{F(s')}.$$

A flow is referred to as a *Markovian flow* if its corresponding probability measure satisfies $P(s \to s' \mid \tau) = P(s \to s' \mid s)$ for any consecutive states $s, s'$ and partial trajectory $\tau = (s_0, \ldots, s)$ ending in $s$. For a Markovian flow and complete trajectory $\tau \in \mathcal{T}$ we have,

$$P(\tau) = \prod_{t=0}^{T} P_F(s_{t+1} \mid s_t) = \prod_{t=0}^{T} P_B(s_t \mid s_{t+1}).$$

GFNs parameterize a Markovian flow on a DAG by modeling the forward transition probabilities $P_F(s' \mid s; \phi)$ together with a normalizing constant $Z_\psi$, which can be interpreted as an approximation to the total amount of flow. The trajectory flow is given by

$$F(\tau; \phi, \psi) = Z_\psi \prod_{t=0}^{T} P_F(s_{t+1} \mid s_t; \phi) = \frac{\prod_{t=0}^{T} F(s_t \to s_{t+1}; \phi, \psi)}{\prod_{t=0}^{T-1} F(s_{t+1}; \phi, \psi)} = Z_\psi \prod_{t=0}^{T} P_B(s_t \mid s_{t+1}; \phi).$$

#### 2.1.2 Training Generative Flow Networks

For a reward function $R$, we want to find transition probabilities $P_F$ and $P_B$, i.e. parameters $\phi$, such that $P_B(s_T \mid s_f; \phi) = R(s_T)/Z =: \pi_T(s_T)$. In some scenarios we want to fix the backward transition model, e.g. a uniform distribution model can be advantageous for exploration, or parameterize it with a distinct set of parameters $\theta$. In this case, the forward and backward transition probabilities do not correspond to the same flow and, under slight overload of notation, we refer to $P_B(s \mid s'; \theta)$ as the backward transition probabilities.

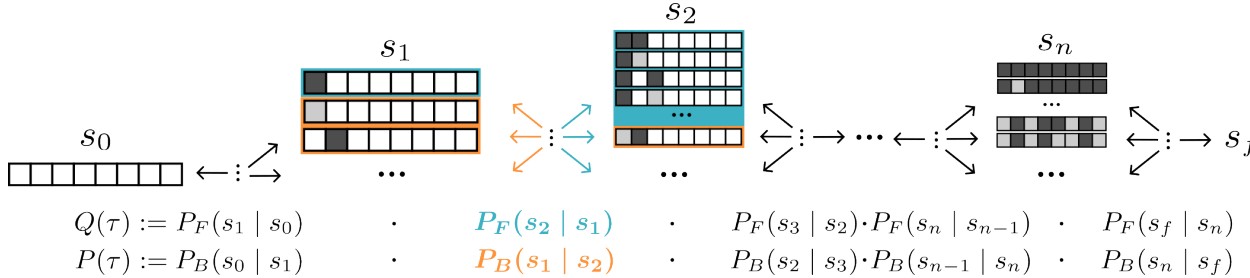

Figure 1: Graph representing the forward- and backward transition model of a GFN over states $s \in \{\emptyset, 0, 1\}^n$ ($\emptyset$ - white, 0 - light gray, 1 - dark gray); the graph is a DAG in the forward- and backward direction respectively. Each forward transition sets one of the $\emptyset$-bits to either 0 or 1 (example in blue), whereas each backward transition deletes one set bit (example in orange).

Bengio et al. (2021a) originally proposed objectives to train GFNs based on the flow matching conditions and a detailed balance condition. Malkin et al. (2022a) observe that optimizing these may lead to inefficient credit propagation to early transitions, especially for long trajectories. To alleviate this, Malkin et al. (2022a) propose an alternative TB objective for complete trajectories

$$\mathcal{L}_{\text{TB}}(\tau, \lambda) = \left(\log \frac{Z_\psi \prod_{t=0}^{T} P_F(s_{t+1}|s_t; \phi)}{R(s_T) \prod_{t=0}^{T-1} P_B(s_t|s_{t+1}; \theta)}\right)^2 = \left(\log \frac{Z_\psi Q(\tau; \phi)}{Z P(\tau; \theta)}\right)^2, \tag{1}$$

where $\lambda = (\phi, \theta, \psi)$ and we define

$$P(\tau; \theta) := \frac{R(s_T)}{Z} \prod_{t=0}^{T-1} P_B(s_t \mid s_{t+1}; \theta), \qquad\qquad Q(\tau; \phi) := \prod_{t=0}^{T} P_F(s_{t+1} \mid s_t; \phi).$$

Trajectories $\tau$ are sampled from a proposal distribution $q$ with full support over the space of trajectories $\mathcal{T}$. The TB objective is optimized using stochastic gradient descent. The gradient w.r.t. all parameters $\lambda = (\phi, \theta, \psi)$ is computed as the average over a batch of $S$ i.i.d. samples. Solutions correspond to fixed points of the (negative) expected gradient

$$\mathbb{E}_{\tau \sim q}\left[\frac{d}{d\lambda}\mathcal{L}_{\text{TB}}(\tau, \lambda)\right] = 0.$$

We can compute an unbiased estimate of this gradient using samples from the proposal distribution,

$$g_{\text{TB}}(\lambda) := \frac{1}{S}\sum_{s=1}^{S}\frac{d}{d\lambda}\mathcal{L}_{\text{TB}}(\tau_s, \lambda), \qquad\qquad \tau_s \sim q.$$

In section 3, we show how optimizing GFNs using the TB objective corresponds to variational inference on complete trajectories. Going forward, we refer to the probability mass functions $Q(\tau; \phi)$ and $P(\tau; \theta)$ over complete trajectories as forward and backward model, respectively.

## 2.2 Variational Inference

The problem of finding corresponding forward and backward transition probabilities can alternatively be phrased as a variational inference problem. The goal is to find parameters $\phi$ and $\theta$ such that the difference between the forward and backward transition probabilities, measured by a suitable divergence, is minimized. Two commonly used divergence measures are the forward Kullback-Leibler divergence (FKL) and reverse Kullback-Leibler divergence (RKL),

$$\mathcal{L}_{\text{RKL}}(\phi, \theta) := \text{KL}(Q(\cdot\,; \phi) \mid P(\cdot\,; \theta)) = \mathbb{E}_{\tau \sim Q}\left[\log \frac{Q(\tau; \phi)}{P(\tau; \theta)}\right] = \mathbb{E}_{\tau \sim Q}\left[-\log w\right], \tag{2}$$

$$\mathcal{L}_{\text{FKL}}(\phi, \theta) := \text{KL}(P(\cdot\,; \theta) \mid Q(\cdot\,; \phi)) = \mathbb{E}_{\tau \sim P}\left[\log \frac{P(\tau; \theta)}{Q(\tau; \phi)}\right] = \mathbb{E}_{\tau \sim P}\left[\log w\right], \tag{3}$$

with the importance weights $w := P(\tau; \theta)/Q(\tau; \phi)$. The divergences can be optimized using stochastic gradient descent with gradients estimated from samples from the forward model $Q$ and backward model $P$, respectively. In most setting, samples from $P$ are not readily available and one has to resort other techniques to generate approximate samples, e.g. using importance sampling or MCMC (Naesseth et al., 2020).

Computing the derivative of $\mathcal{L}_{\text{RKL}}$ w.r.t. parameters $\theta$ of the backward transition model is straightforward, the dependence only appears in the log-weights. We can approximate the resulting expected gradient using $S$ samples from the forward model,

$$\frac{d}{d\theta} \mathcal{L}_{\text{RKL}}(\phi, \theta) = \mathbb{E}_{\tau \sim Q} \left[ -\frac{d}{d\theta} \log P(\tau; \theta) \right] \approx g_{\text{RKL}}^{\theta}(\phi, \theta) := \frac{1}{S} \sum_{s=1}^{S} -\frac{d}{d\theta} \log P(\tau_s; \theta), \qquad \tau_s \sim Q(\cdot; \phi).$$

Similarly, the derivative of $\mathcal{L}_{\text{FKL}}$ w.r.t. parameters $\phi$ of the forward transition model and corresponding gradient estimator $g_{\text{RKL}}^{\phi}$ are

$$\frac{d}{d\phi} \mathcal{L}_{\text{FKL}}(\phi, \theta) = \mathbb{E}_{\tau \sim P} \left[ -\frac{d}{d\phi} \log Q(\tau; \phi) \right] \approx g_{\text{FKL}}^{\phi}(\phi, \theta) := \frac{1}{S} \sum_{s=1}^{S} -\frac{d}{d\phi} \log Q(\tau_s; \phi), \qquad \tau_s \sim P(\cdot; \theta).$$

Computing derivative of $\mathcal{L}_{\text{RKL}}$ w.r.t. $\phi$ and derivative of $\mathcal{L}_{\text{FKL}}$ w.r.t. $\theta$ on the other hand involves computing a so-called score-function gradient,

$$\begin{aligned}
\frac{d}{d\phi} \mathcal{L}_{\text{RKL}}(\phi, \theta) &= \sum_{\tau \in \mathcal{T}} \log \frac{Q(\tau; \phi)}{P(\tau; \theta)} \frac{d}{d\phi} Q(\tau; \phi) + \left( \frac{d}{d\phi} \log \frac{Q(\tau; \phi)}{P(\tau; \theta)} \right) Q(\tau; \phi) \\
&= \sum_{\tau \in \mathcal{T}} \log \frac{Q(\tau; \phi)}{P(\tau; \theta)} Q(\tau; \phi) \frac{d}{d\phi} \log Q(\tau; \phi) + Q(\tau; \phi) \frac{d}{d\phi} \log Q(\tau; \phi) \\
&= \mathbb{E}_{\tau \sim Q} \left[ (-\log w + 1) \frac{d}{d\phi} \log Q(\tau; \phi) \right] = \mathbb{E}_{\tau \sim Q} \left[ -\log w \frac{d}{d\phi} \log Q(\tau; \phi) \right]
\end{aligned}$$

Importantly, we can cancel-out the additional score-function term (last equality of above equation) as $\mathbb{E}_{\tau \sim Q}[a \frac{d}{d\phi} \log Q(\tau; \phi)] = 0$ for any constant $a$. The corresponding score-function gradient estimator is thus

$$g_{\text{RKL}}^{\phi}(\phi, \theta) := \frac{1}{S} \sum_{s=1}^{S} -\log w_s \frac{d}{d\phi} \log Q(\tau_s; \phi), \qquad w_s := \frac{P(\tau_s; \theta)}{Q(\tau_s; \phi)}, \qquad \tau_s \sim Q(\cdot; \phi).$$

Analogously, we can compute a score function gradient of $\mathcal{L}_{\text{FKL}}$ w.r.t. $\theta$ and corresponding estimator

$$\mathbb{E}_{\tau \sim P} \left[ \log w \frac{d}{d\theta} \log P(\tau; \theta) \right] \approx g_{\text{FKL}}^{\theta}(\phi, \theta) := \frac{1}{S} \sum_{s=1}^{S} \log w_s \frac{d}{d\theta} \log P(\tau_s; \theta), \qquad \tau_s \sim P(\cdot; \theta).$$

Score-function gradient estimators can exhibit high variance (Ranganath et al., 2013), which can be problematic for learning variational approximations via stochastic gradient descent, and hence it is often essential to employ variance reduction techniques.

### 2.2.1 Variance reduction techniques for score-function estimators

A commonly used technique to reduce the variance of score-function estimators is to use a control variate $h$ (Ross, 1997) to replace the gradient estimator $g$ with the modified estimator $g' = g + c(h - \mathbb{E}[h])$, where $c$ is a scaling parameter. Control variates leave the expected value of the gradient estimator $g$ unchanged, $\mathbb{E}[g] = \mathbb{E}[g']$, but has the potential to reduce the variance. Indeed, for a given control variate $h$ we can minimize the variance of $g'$

$$\text{Var}[g'] = \text{Var}[g] + c^2 \text{Var}[h] + 2c \text{Cov}[g, h] \tag{4}$$

with respect to the scaling $c$. The scaling that minimizes the variance is

$$c^* = \arg\min_c \mathrm{Var}[g'] = -\frac{\mathrm{Cov}[g,h]}{\mathrm{Var}[h]}.$$

When $g$, and hence $g'$, is a vector, we are typically interested in the scalar- or vector-valued scaling $c$ that minimizes the trace of the covariance matrix of $g'$. Note that the trace obtained with the optimal vector-valued scaling is a lower bound on the trace obtained with the optimal scalar-valued scaling as

$$\min_{c^{\mathrm{vec}}} \sum_{d=1}^{D} \mathrm{Var}[g'_d] = \sum_{d=1}^{D} \min_{c^{\mathrm{vec}}_d} \mathrm{Var}[g'_d] \le \min_{c^{\mathrm{sca}}} \sum_{d=1}^{D} \mathrm{Var}[g'_d].$$

The optimal scalar- and vector-valued scalings are

$$c^*_{\mathrm{sca}} = -\frac{\sum_{d=1}^{D} \mathrm{Cov}[g_d,h_d]}{\sum_{d=1}^{D} \mathrm{Var}[h_d]}, \qquad\qquad c^*_{\mathrm{vec,d}} = -\frac{\mathrm{Cov}[g_d,h_d]}{\mathrm{Var}[h_d]}.$$

The score function $\frac{d}{d\phi}\log Q(\tau;\phi)$ (Ranganath et al., 2013) is a useful and easy to compute control variate when optimizing the reverse KL divergence, which we will use as our running example. Using the score function as a control variate simplifies the expression of the optimal baseline [1] and the resulting gradient estimator such that the scaling $c$ can simply be added to the (negative) log-importance weight,

$$g' = \underbrace{\frac{1}{S}\sum_{i=s}^{S} -\log w_s \,\frac{d}{d\phi}\log Q(\tau_s;\phi)}_{g} + c\Big(\frac{d}{d\phi}\log Q(\tau_s;\phi) - \underbrace{\mathbb{E}\left[\frac{d}{d\phi}\log Q(\tau;\phi)\right]}_{=0}\Big)$$

$$= \frac{1}{S}\sum_{i=s}^{S}\big(-\log w_s + c\big)\frac{d}{d\phi}\log Q(\tau_s;\phi).$$

**Monte Carlo Estimation.** We can estimate the optimal scaling with the same $S$ i.i.d. samples $\tau_s \sim Q(\tau_s;\theta)$ used to estimate $g$. However, in order for the gradient estimator to remain unbiased, we have to employ a *leave-one-out* (LOO) estimator $\hat{c}_s$ (Mnih and Rezende, 2016), which only makes use of samples $\{\hat{\tau}_{s'} \mid s' \ne s\}$, such that

$$\mathbb{E}\left[\frac{1}{S}\sum_{i=s}^{S}(-\log w_s + \hat{c}_s)\frac{d}{d\phi}\log Q(\tau_s;\phi)\right] = \mathbb{E}\left[-\log w_s \frac{d}{d\phi}\log Q(\tau_s;\phi)\right].$$

The leave-on-out estimate of the optimal scaling for the $d$-th dimension of $c^*$ is

$$\hat{c}^*_{d,s} = -\frac{\widehat{\mathrm{Cov}}_s[g_d,h_d]}{\widehat{\mathrm{Var}}_s[h_d]},$$

where $\widehat{\mathrm{Cov}}_s[\cdot,\cdot]$, and $\widehat{\mathrm{Var}}_s[\cdot]$ are empirical LOO covariance and variance estimates, respectively. Note that estimating the optimal scaling requires access to per-sample gradients and hence requires $S$ forward-backward passes on the computations graph in many reverse-mode automatic differentiation frameworks[2]. Two popular non-optimal scaling choices that are easily computed and do not require access to gradient information are $c^{\log w} = \mathbb{E}[\log w]$ and $c^{\log Z} = \log\mathbb{E}[w]$ with corresponding LOO estimators

$$\hat{c}^{\log w}_s := \frac{1}{S-1}\sum_{s'=1,s'\ne s}^{S}\log w_{s'} \qquad\qquad \hat{c}^{\log Z}_s := \log\frac{1}{S-1}\sum_{s'=1,s'\ne s}^{S} w_{s'}.$$

---

[1]The expressions for the optimal baseline simplifies due to fact that the expectation of the score function is zero.

[2]We used jax (Bradbury et al., 2018) to efficiently compute per-sample gradients to estimate the optimal control variates.

Interestingly, for $\hat{c}^{\log w}$ one can show that it is sufficient to only compute the fixed scaling $\hat{c}^{\log w}$ and instead correct by a factor $\frac{S-1}{S}$ to obtain an unbiased estimate of $g'$ ,

$$\frac{1}{S}\sum_{i=1}^{S}\big(-\log w_s + \hat{c}_s^{\log w}\big)\frac{d}{d\phi}\log Q(\tau_s;\phi) = \frac{1}{S-1}\sum_{i=1}^{S}\big(-\log w_s + \underbrace{\frac{1}{S}\sum_{j=1}^{S}\log w_j}_{\hat{c}^{\log w}}\big)\frac{d}{d\phi}\log Q(\tau_s;\phi).$$

In Section 3 we show how we can leverage these variance reduction techniques for training GFNs by identifying scenarios in which training GFNs with the TB objective is equivalent to performing variational inference with a score-function gradient estimator.

## 3 Variational Inference for Generative Flow Networks

The trajectory balance objective and variational objectives, introduced in Section 2.2.1, all try to find a forward model $Q$ and backward model $P$ such that

$$P(\tau;\theta) = \pi_T(s_T)\prod_{t=0}^{T-1}P_B(s_t \mid s_{t+1};\theta) \approx \prod_{t=0}^{T}P_F(s_{t+1} \mid s_t;\phi) = Q(\tau;\phi),$$

and hence terminating states $s_T$ which are approximately distributed according to $\pi_T$, which is proportional to the reward $R$. While the TB objective can be optimized with samples from any proposal distribution that has full support on $\mathcal{T}$, it is commonly optimized with samples from either the forward model $\tau_F \sim Q$ or the backward model $\tau_B \sim P$. Similarly, variational inference commonly optimizes the RKL divergence or FKL divergence, which can be estimated by sampling from the forward model and backward model, respectively.

Zhang et al. (2022b) propose a special case of the trajectory balance objective using a proposal that first samples a Bernoulli random variable $u \sim \mathcal{B}(\alpha)$. This variable then determines whether the trajectory samples are drawn from the forward model or the backward model. The corresponding expected gradient is

$$\mathbb{E}_{u\sim\mathcal{B}(\alpha)}\left[[u=0]\mathbb{E}_{\tau\sim P(\cdot;\theta)}\left[\frac{d}{d\lambda}\mathcal{L}_{\text{TB}}(\tau,\lambda)\right] + [u=1]\mathbb{E}_{\tau\sim Q(\cdot;\phi)}\left[\frac{d}{d\lambda}\mathcal{L}_{\text{TB}}(\tau,\lambda)\right]\right] \quad (5)$$

$$=\alpha\mathbb{E}_{\tau_B\sim P(\cdot;\theta)}\left[\frac{d}{d\lambda}\mathcal{L}_{\text{TB}}(\tau,\lambda)\right] + (1-\alpha)\mathbb{E}_{\tau\sim Q(\cdot;\phi)}\left[\frac{d}{d\lambda}\mathcal{L}_{\text{TB}}(\tau,\lambda)\right]. \quad (6)$$

We can approximate the expected gradient by approximating the expectation w.r.t. the forward and backward model for any backward ratio $\alpha \in [0,1]$, which is equivalent to optimizing a weighted sum of TB objectives,

$$\mathcal{L}_{\alpha\text{TB}}(\tau_F,\tau_B,\lambda) := \alpha\mathcal{L}_{\text{TB}}(\tau_B,\lambda) + (1-\alpha)\mathcal{L}_{\text{TB}}(\tau_F,\lambda),$$

where $\tau_F \sim Q(\cdot;\phi)$ and $\tau_B \sim P(\cdot;\theta)$. We can similarly define a convex combination of the two KL divergences, which penalizes the FKL objective and RKL objective with $\alpha$ and $(1-\alpha)$, respectively,

$$\mathcal{L}_{\alpha\text{KL}}(\phi,\theta,\alpha) =\alpha\mathcal{L}_{\text{FKL}}(\phi,\theta). + (1-\alpha)\mathcal{L}_{\text{RKL}}(\phi,\theta)$$

Like RKL and FKL, this is a divergence which is non-negative and zero if and only if $P = Q$. For $\alpha = 0.5$ the objective recovers the Jeffreys divergence.

We are now equipped to compare the various objectives for different settings of $\alpha$ and different parameterizations of the forward and backward model. Specifically, we differentiate between two settings: (1) $P_F$ and $P_B$ (and hence $Q$ and $P$) have distinct parameters $\phi$ and $\theta$ respectively, and (2) $P_F$ and $P_B$ share parameters $\eta = \phi = \theta$. The expected gradient of $\mathcal{L}_{\alpha\text{TB}}$ can be computed as the convex combination of the expected gradient of $\mathcal{L}_{\text{TB}}$ w.r.t. samples from the forward model and backward model (see Equation 5). Similarly, $\mathcal{L}_{\alpha\text{KL}}$ can be computed as convex combination of $\mathcal{L}_{\text{RKL}}$ and $\mathcal{L}_{\text{FKL}}$. Thus, in the following we study the cases $\alpha = 0$ and $\alpha = 1$ separately and results for $0 < \alpha < 1$ follow accordingly.

### 3.1 Forward model and backward model with shared parameters

If the forward and backward model share parameters $\eta = (\phi, \theta)$, e.g. when they are parameterized by the same GFN, the expected gradient of the TB objective (Equation 1) takes the form

$$\mathbb{E}_{\tau \sim q(\cdot;\eta)}\left[\frac{d}{d\lambda}\mathcal{L}_{\mathrm{TB}}(\tau, \lambda)\right] = -2\mathbb{E}_{\tau \sim q(\cdot;\eta)}\left[\left(\log w + \log \frac{Z}{Z_\psi}\right)\left(\frac{d}{d\psi}\log Z_\psi + \frac{d}{d\eta}\log Q(\tau;\eta) - \frac{d}{d\eta}\log P(\tau;\eta)\right)\right],$$

where the proposal $q$ is either the forward model $Q(\tau; \phi)$ ($\alpha = 0$) or backward model $P(\tau; \theta)$ ($\alpha = 1$). The corresponding gradients of the RKL and FKL divergences are

$$\frac{d}{d\eta}\mathcal{L}_{\mathrm{RKL}}(\eta) = -\mathbb{E}_{\tau \sim Q(\cdot;\eta)}\left[(\log w + c)\frac{d}{d\eta}\log Q(\tau;\eta) + \frac{d}{d\eta}\log P(\tau;\eta)\right],$$

$$\frac{d}{d\eta}\mathcal{L}_{\mathrm{FKL}}(\eta) = \mathbb{E}_{\tau \sim P(\cdot;\eta)}\left[(\log w + c)\frac{d}{d\eta}\log P(\tau;\eta) - \frac{d}{d\eta}\log Q(\tau;\eta)\right],$$

where $c$ is a scaling parameter as discussed in Section 2.2.1.

### 3.2 Forward model and backward model with distinct parameters

**Sampling from the forward model ($\alpha = 0$).** In the case where we are using samples from the forward model $\tau \sim Q(\cdot; \phi)$ only, the expected TB gradients reduce to

$$\mathbb{E}_{\tau \sim Q(\cdot;\phi)}\left[\frac{d}{d\phi}\mathcal{L}_{\mathrm{TB}}(\tau, \lambda)\right] = -2\mathbb{E}_{\tau \sim Q(\cdot;\phi)}\left[\left(\log w + \log \frac{Z}{Z_\psi}\right)\frac{d}{d\phi}\log Q(\tau;\phi)\right],$$

$$\mathbb{E}_{\tau \sim Q(\cdot;\phi)}\left[\frac{d}{d\theta}\mathcal{L}_{\mathrm{TB}}(\tau, \lambda)\right] = 2\mathbb{E}_{\tau \sim Q(\cdot;\phi)}\left[\left(\log w + \log \frac{Z}{Z_\psi}\right)\frac{d}{d\theta}\log P(\tau;\theta)\right],$$

$$\mathbb{E}_{\tau \sim Q(\cdot;\phi)}\left[\frac{d}{d\psi}\mathcal{L}_{\mathrm{TB}}(\tau, \lambda)\right] = 2\mathbb{E}_{\tau \sim Q(\cdot;\phi)}\left[\left(\log w + \log \frac{Z}{Z_\psi}\right)\frac{d}{d\psi}\log Z_\psi\right].$$

Interestingly, the expected gradient w.r.t. $\phi$ does not depend on $\log Z_\psi$ and is proportional to the gradient of the standard score-function gradient for the reverse KL-divergence

$$\frac{d}{d\phi}\mathcal{L}_{\mathrm{RKL}}(\phi, \theta) = -\mathbb{E}_{\tau \sim Q(\cdot;\phi)}\left[(\log w + c)\frac{d}{d\phi}\log Q(\tau;\phi)\right] = \frac{1}{2}\mathbb{E}_{\tau \sim Q(\cdot;\phi)}\left[\frac{d}{d\phi}\mathcal{L}_{\mathrm{TB}}(\tau, \lambda)\right].$$

Hence, solutions of the corresponding optimization problem correspond to fixed points of the (negative) expected gradient. Moreover, the term $\log Z/Z_\psi$ can be interpreted as a learned scaling parameter $c_\psi$ for

| | $\alpha = 0, \quad \tau \sim Q(\cdot;\phi)$ | $\alpha = 1, \quad \tau \sim P(\cdot;\theta)$ |
|---|---|---|
| $\frac{d}{d\phi}\mathcal{L}_{\mathrm{KL}}(\phi, \theta)$ | $-\mathbb{E}\left[\log w \frac{d}{d\phi}\log Q(\tau;\phi)\right]$ | $-\mathbb{E}\left[\frac{d}{d\phi}\log Q(\tau;\phi)\right]$ |
| $\mathbb{E}[\frac{d}{d\phi}\mathcal{L}_{\mathrm{TB}}(\phi, \theta, \psi)]$ | $-2\mathbb{E}\left[\log w \frac{d}{d\phi}\log Q(\tau;\phi)\right]$ | $-2\mathbb{E}\left[\left(\log w + \log \frac{Z}{Z_\psi}\right)\frac{d}{d\phi}\log Q(\tau;\phi)\right]$ |
| $\frac{d}{d\theta}\mathcal{L}_{\mathrm{KL}}(\phi, \theta)$ | $-\mathbb{E}\left[\frac{d}{d\theta}\log P(\tau;\theta)\right]$ | $\mathbb{E}\left[\log w \frac{d}{d\theta}\log P(\tau;\theta)\right]$ |
| $\mathbb{E}[\frac{d}{d\theta}\mathcal{L}_{\mathrm{TB}}(\phi, \theta, \psi)]$ | $2\mathbb{E}\left[\left(\log w + \log \frac{Z}{Z_\psi}\right)\frac{d}{d\theta}\log P(\tau;\theta)\right]$ | $2\mathbb{E}\left[\log w \frac{d}{d\theta}\log P(\tau;\theta)\right]$ |

Table 1: Gradient expressions w.r.t. parameters of the forward model ($\phi$) and backward model ($\theta$) for the $\alpha$TB and $\alpha$KL objective. For $\alpha = 0$, the gradient w.r.t. $\phi$ of the $\alpha$KL objective is proportional to the expected gradient of the $\alpha$TB objective (highlight in first column). Analogously, for $\alpha = 1$, the gradient w.r.t. $\theta$ of the $\alpha$KL objective is proportional to the expected gradient of the $\alpha$TB objective (highlight in second column).

variance reduction similar to the control variates discussed in section 2.2.1. Optimizing the TB objective w.r.t. parameters of the forward model is equivalent to optimizing a RKL divergence using a score-function estimator with a learned scaling parameter $c_\psi$, updated according to the gradient described above. This insight also suggests that the control variate described in Section 2.2.1 can be used as an alternative to the learned baseline to reduce the variance of the expected gradient estimates of the trajectory balance objective.

The expression of the gradient of the RKL w.r.t. parameters of the backward model $\theta$ differs from the expected gradient of the corresponding TB objective

$$\frac{d}{d\theta}\mathcal{L}_{\mathrm{RKL}}(\phi, \theta) = -\mathbb{E}_{\tau \sim Q(\cdot; \phi)}\left[\frac{d}{d\theta}\log P(\tau; \theta)\right].$$

The integrand differs by a multiplicative factor $\log w + c_\psi$.

Intuitively, if the likelihood of a sample is higher under the backward transition model $P$ than under the forward transition model $Q$ by more than predicted by $-c_\psi = \log(Z_\psi/Z)$, then $\log w + c_\psi < 0$ and the TB objective tries to increase the likelihood of the sample under $P$ and vice versa. In contrast, the gradient of the RKL objective tries to always maximize the likelihood of samples under the backward transition model, which achieves its global maximum for $P = Q$. Due to the fact that $\sum_\tau P(\tau; \theta) = 1$, increasing the probability of $P(\tau; \theta)$ for some $\tau$ decreases the probability of other trajectories *indirectly*. Hence, while both objectives have the same global minima for flexible enough $Q$ and $P$, their optimization dynamics may differ.

**Sampling from the backward model ($\alpha = 1$).** When samples are taken from the backward model $\tau \sim P(\cdot; \theta)$ the expected TB gradients reduce to

$$\mathbb{E}_{\tau \sim P(\cdot; \theta)}\left[\frac{d}{d\phi}\mathcal{L}_{\mathrm{TB}}(\tau, \lambda)\right] = -2\mathbb{E}_{\tau \sim P(\cdot; \theta)}\left[\left(\log w + \log\frac{Z}{Z_\psi}\right)\frac{d}{d\phi}\log Q(\tau; \phi)\right],$$

$$\mathbb{E}_{\tau \sim P(\cdot; \theta)}\left[\frac{d}{d\theta}\mathcal{L}_{\mathrm{TB}}(\tau, \lambda)\right] = 2\mathbb{E}_{\tau \sim P(\cdot; \theta)}\left[\left(\log w + \log\frac{\cancel{Z}}{Z_\psi}\right)\frac{d}{d\theta}\log P(\tau; \theta)\right],$$

$$\mathbb{E}_{\tau \sim P(\cdot; \theta)}\left[\frac{d}{d\psi}\mathcal{L}_{\mathrm{TB}}(\tau, \lambda)\right] = 2\mathbb{E}_{\tau \sim P(\cdot; \theta)}\left[\left(\log w + \log\frac{Z}{Z_\psi}\right)\frac{d}{d\psi}\log Z_\psi\right].$$

Here, a similar observation holds. The expected gradient, w.r.t. $\theta$, of the TB objective is proportional to the corresponding gradient of the forward KL-divergence w.r.t. parameters $\theta$

$$\frac{d}{d\theta}\mathcal{L}_{\mathrm{FKL}}(\phi, \theta) = \mathbb{E}_{\tau \sim P(\cdot; \theta)}\left[\log w \frac{d}{d\theta}\log P(\tau; \theta)\right] = \frac{1}{2}\mathbb{E}_{\tau \sim P(\cdot; \theta)}\left[\frac{d}{d\theta}\mathcal{L}_{\mathrm{TB}}(\phi, \theta, \tau)\right].$$

Again, solutions of the corresponding optimization problem correspond to fixed points of the (negative) expected gradient. Moreover, analogously to the previous case, optimizing the TB objective w.r.t. $\theta$ is equivalent to optimizing a FKL divergence w.r.t. $\theta$ using a score-function estimator with a learned scaling parameter $c_\psi$.

The expression of the gradient of the FKL w.r.t. parameters of the forward model $\phi$ analogously differs from the expected gradient of the corresponding TB objective by a factor $\log w + c_\psi$ in the integrand,

$$\frac{d}{d\phi}\mathcal{L}_{\mathrm{FKL}}(\phi, \theta) = -\mathbb{E}_{\tau \sim P(\cdot; \theta)}\left[\frac{d}{d\phi}\log Q(\tau; \phi)\right].$$

Observing the expected gradients of the TB objective and corresponding gradients of the RKL and FKL shows that in certain cases optimizing the TB objective is equivalent to variational inference using reverse or forward KL divergences. This observation also suggests that we can leverage the various variance reduction techniques for score-function estimators developed in the variational inference literature. We summarize the gradients w.r.t. the parameters of the forward- and backward model in Table 1.

## 4 Experiments

We have shown that for certain settings, optimizing the $\alpha$TB objective is equivalent to optimizing the $\alpha$KL objective, in the sense that the fixed points are the same and the expected gradient of the $\alpha$TB objective is proportional to the gradient of the $\alpha$KL objective. In these settings we can use the variance reduction techniques for score-function gradient estimators to reduce the variance of the expected gradients of the TB objective. In settings where optimizing the $\alpha$TB objective and $\alpha$KL objective is not equivalent, it is not immediately clear if optimizing the $\alpha$KL objective is advantageous over optimizing the $\alpha$TB objective, or vice versa. In the following we compare the performance of the $\alpha$TB and $\alpha$KL objective with different control variates and different backward ratios $\alpha$.

**Generating samples from the target density.** Optimizing the $\alpha$TB and $\alpha$KL objective with $\alpha > 0$ require samples from the ground truth distribution $s_T = x \sim \pi_{\text{GT}}$. In these setting we require either (1) access to online samples $x \sim \pi_{\text{GT}}$ (e.g. in the synthetic density experiment in 4.1) or (2) access to a data set $\mathcal{X} = \{x_i\}_{i=1}^n$ containing samples $x_i \sim \pi_{GT}$ which can be uniformly subsampled during optimization.

**Evaluation metrics.** If samples from the ground-truth target distribution are available we can sample trajectories from the backward model conditioned on $x$. Let

$$P_B(s_{0:T-1} \mid s_T; \theta) := \prod_{t=0}^{T-1} P_B(s_t \mid s_{t+1}; \theta) \quad \text{and} \quad P_F(s_{1:T-1}|s_0; \phi) := \prod_{t=0}^{T-2} P_F(s_{t+1}|s_t; \phi).$$

Then, we can estimate the marginal likelihood of the data under the the forward model using importance sampling,

$$\frac{1}{N} \sum_{i=1}^{N} \frac{P_F(x|s_{T-1}^i; \phi) P_F(s_{1:T-1}^i|s_0^i; \phi)}{P_B(s_{0:T-1}^i \mid x; \theta)}, \qquad s_{0:T-1}^i \sim P_B(s_{0:T-1} \mid x; \theta). \tag{7}$$

If no data is available we will report the expected log-weight $\mathbb{E}_{\tau \sim Q(\cdot; \phi)}[\log w] \leq \log Z$.

**Structure and representation of the state space.** Following Zhang et al. (2022b) we target a discrete distribution over terminating states on $\mathcal{S}_T = \{0, 1\}^D$ by consecutively sampling values in $\{0, 1\}$ for each step. To this end we define the state space $\mathcal{S} = \{\emptyset, 0, 1\}^D \cup \{s_f\}$, where $\emptyset$ indicates that no bit value has been sampled for the corresponding position yet. We further define edges

$$E = \{(s, s') : s \in \mathcal{S} \setminus \{s_f\} \wedge s' \in \mathcal{S}'(s)\} \cup \{(s, s_f) : s \in \mathcal{S}_T\}, \quad \mathcal{S}'(s) = \{s' \in \mathcal{S} \setminus \{s_f\} : |s| = |s'| - 1\},$$

where $|s|$ denotes the number of set bits in $s$. With these definitions in place we define a DAG $G(\mathcal{S}, E)$ that specifies the structure of the state space (see Figure 1). For mathematical convenience, we map the states $s$ to numeric representations $\tilde{s}$ in which $\emptyset$, 0 and 1 are replaced by 0, $-1$ and 1 respectively. This allows us to compute the number of set bits $|s| = \sum_d |\tilde{s}_d|$, and the location and type of the bit added by a transition $s \to s'$ as the signed one-hot vector $\tilde{s}' - \tilde{s}$. We can also compute the state $\neg \tilde{s}'(s, s') = \tilde{s} - (\tilde{s}' - \tilde{s})$ that results from flipping the newly added bit in $s'$. These operations are useful for defining the transition model.

**Transition model.** We consider a fixed backward transition model $P_B(s_t \mid s_{t+1})$ which uniformly at random select a set bit and replaces it with $\emptyset$. The forward transition model $P_F(s_{t+1} \mid s_t; \phi)$ uniformly at random selects a $\emptyset$-bit and and replaces it with a bit value sampled from a Bernoulli distribution, whose parameters (logits) are the output of a function $f_\phi : \mathcal{S} \times \mathcal{S} \to \mathbb{R}_+$. The corresponding probability mass functions of the forward- and backward transition model are

$$P_B(s_t \mid s_{t+1}) = \frac{1}{|s_{t+1}|}, \qquad P_F(s_{t+1}|s_t; \phi) = \frac{1}{D - |s_t|} \frac{f_\phi(\tilde{s}_t')}{f_\phi(\tilde{s}_t') + f_\phi(\neg \tilde{s}'(s_t, s_{t+1}))}.$$

In practice $f_\phi : \mathbb{R}^D \to \mathbb{R}^{D \times 2}$ is a vector valued function parameterized by a Multilayer Perceptron (MLP) with weights $\phi$. Given a state $s_t$, it produces $D$ pairs of logits associated with positions in the state vector. The uniformly drawn position $d$ of the added bit is then used to select the corresponding logits $f_\phi(s)_d \in \mathbb{R}^2$.

## 4.1 Synthetic densities

To model a discrete ground-truth distribution $\pi_{\mathrm{GT}}$ over terminating states we follow Dai et al. (2020); Zhang et al. (2022b) and discretize a continuous distribution $\pi_{\mathrm{GT}}^{\mathrm{cont}} : \mathbb{R}^2 \to \mathbb{R}^+$ into $2^{16}$ equally sized grid cells along each dimension. The cells are remapped to Gray code such that neighbouring grid cells differ in exactly one bit and the resulting pair of 16-bit vectors is concatenated to obtain a single 32-bit vector.

We are interested in two settings: (1) Learning a forward model $Q(\tau; \phi)$ such that its marginal distribution $Q_T(s_T; \phi)$ approximates a fixed distribution $\pi_T(s_T) = \pi_{\mathrm{GT}}(s_T)$ over terminating states, and (2) learning a forward model and energy function $\xi : \{0, 1\}^{32} \to \mathbb{R}$ jointly such that $\pi_T(s_T; \theta) \propto \exp(-\xi(s_T, \theta)) \approx \pi_{\mathrm{GT}}(s_T)$. We optimize the energy function by minimizing the negative log-likelihood via stochastic gradient descent, interleaving gradient updates to the forward model and energy function.

We approximate the gradient of the log-marginal likelihood

$$-\frac{d}{d\theta} \log \pi_T(s_T; \theta) = \frac{d}{d\theta} \left( \xi(s_T; \theta) + \log Z_\theta \right) = \frac{d}{d\theta} \xi(s_T; \theta) - \mathbb{E}_{s_T \sim \pi_T(\cdot; \theta)} \left[ \xi(s_T; \theta) \right]$$

using a contrastive divergence-based approach (Hinton, 2002), which replaces the expectation w.r.t. $\pi_T$ with an expectation w.r.t. the marginal distribution of a $K$-step Metropolis-Hastings (MH) chain $m(x' \mid x)$ initialized at data $x$,

$$\mathbb{E}_{x \sim \mathcal{U}(\mathcal{X})} \left[ \frac{d}{d\theta} \xi(x; \theta) - \mathbb{E}_{x' \sim m(x' \mid x)} \left[ \xi(x'; \theta) \right] \right].$$

The MH updates uses the GFN to construct proposals (Zhang et al., 2022b). For $K \to \infty$ this gradient update recovers the expected gradient of the log-marginal likelihood.

We evaluate the $\alpha$TB objective and $\alpha$KL objective with a learned control variate for different backward ratios $\alpha$. For each backward ratio we consider two settings: (1) jointly learning the energy function $\xi$ and parameters of the GFN, and (2) using a previously learned fixed energy and learning parameters only. We report the negative log-likelihood in Table 2). All numbers are averages over 10 independently trained models.

We find that, unsurprisingly, for $\alpha = 0$, in which case optimizing the $\alpha$TB objective is equivalent to optimizing the $\alpha$KL objective with a learned control variate both objectives perform comparably, i.e. their average negative log-likelihoods are withing one standard deviation of each other (see Table 2). Similarly, for $0 < \alpha < 1$, both objectives perform similarly, with $\alpha$TB having a slight edge over $\alpha$KL in terms of negative log-likelihood. For $\alpha = 1$, i.e. when sampling from backward model only, the performance of $\alpha$TB drops significantly while the performance of the $\alpha$KL objective remains stable.

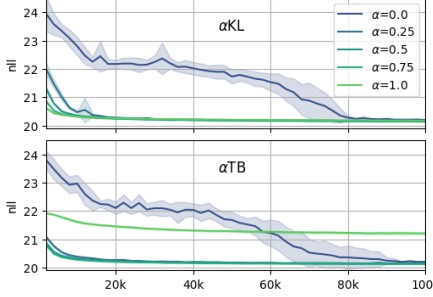

Figure 2: Negative log-likelihood over training on 2spirals for fixed energy $\xi$ and different backward ratios $\alpha$.

| method | 2spirals learned $\xi$ | 8gaussians learned $\xi$ | 2spirals fixed $\xi$ | 8gaussians fixed $\xi$ |
|---|---|---|---|---|
| $\alpha$TB, $\alpha=0.0$ | 20.163±0.013 | 20.006±0.015 | 20.201±0.032 | 20.038±0.014 |
| $\alpha$TB, $\alpha=0.25$ | 20.133±0.010 | 20.000±0.011 | 20.149±0.015 | 20.002±0.015 |
| $\alpha$TB, $\alpha=0.5$ | 20.119±0.006 | **19.997**±0.010 | 20.134±0.014 | 19.999±0.014 |
| $\alpha$TB, $\alpha=0.75$ | **20.118**±0.008 | 20.002±0.008 | **20.126**±0.008 | **19.995**±0.010 |
| $\alpha$TB, $\alpha=1.0$ | 20.994±0.037 | 20.088±0.008 | 21.207±0.051 | 20.076±0.012 |
| $\alpha$KL, $\alpha=0.0$ | 20.171±0.015 | 20.021±0.018 | 20.196±0.017 | 20.045±0.015 |
| $\alpha$KL, $\alpha=0.25$ | 20.142±0.012 | 19.999±0.007 | 20.147±0.009 | 20.003±0.008 |
| $\alpha$KL, $\alpha=0.5$ | 20.145±0.008 | 20.003±0.014 | 20.144±0.005 | 20.006±0.010 |
| $\alpha$KL, $\alpha=0.75$ | 20.160±0.008 | 20.019±0.009 | 20.152±0.010 | 20.011±0.011 |
| $\alpha$KL, $\alpha=1.0$ | 20.174±0.009 | 20.019±0.010 | 20.174±0.010 | 20.016±0.008 |

Table 2: Negative log-likelihood of test data under learned GFN policy with learned control variate for learned and fixed energy functions and different backward ratios $\alpha$.

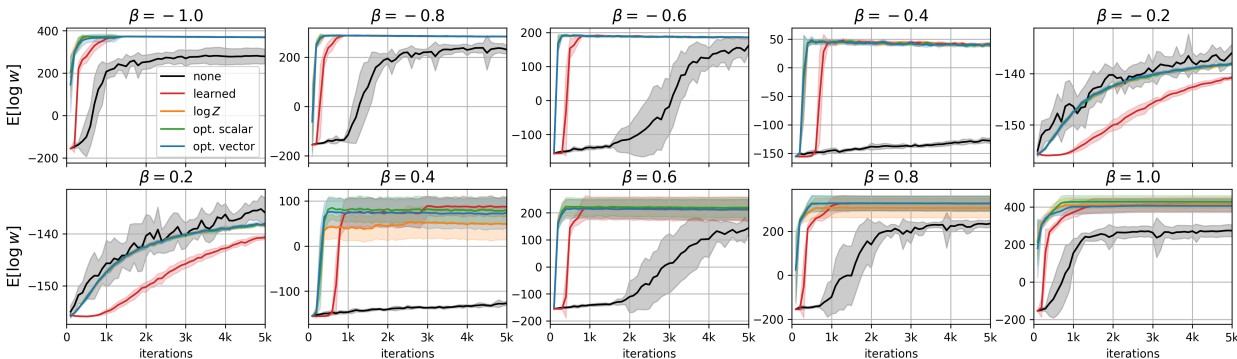

Figure 3: Expected log importance-weight over training. Using the estimated baselines leads to faster convergence during early stages of training.

## 4.2 Ising model

We are modeling a discrete distribution over terminating states $s_T \in \{-1, 1\}^D$ corresponding to the grid cells of an Ising model,

$$\pi_T(s_T) \propto \exp(-\beta H(s_T)), \qquad H(s_T) = -\frac{1}{2}s_T^\top A_N s_T, \qquad (8)$$

where $A$ is the adjacency matrix of a $N \times N$ ($D = N^2$) grid with periodic boundary conditions, and $\beta$ is the interaction strength.

As we do not have access to ground truth samples from the Ising model, we are training the GFN with $\alpha = 0$. In this setting, optimizing the $\alpha$TB objective and $\alpha$KL objective is equivalent and hence we focus on the effect of replacing the learned control variate, that is used in the original $\alpha$TB objective, with estimated control variate typically used to reduce the variance in score-function estimators.

We report the expected log-weights for different control variates and values of $\beta$ averaged over 10 independently trained GFNs in Table 3, and show samples from a GFN and samples generated by running a MH chain for qualitative comparison in Figure 4. For computational efficiency we compute a slightly biased global estimate for the optimal control variates instead of an unbiased LOO estimate, which we have found to not noticeable influence the result. While using the estimated control variates helps the GFN to converge faster, specifically in early stages of training (see Figue 3), we find that it has no significant influence on the final performance.

| method | control variate | $\beta = -1.0$ | $\beta = -0.8$ | $\beta = -0.6$ | $\beta = -0.4$ | $-0.2$ |
|---|---|---|---|---|---|---|
| $\alpha$KL, $\alpha$=0.0 | none | $279.54 \pm 34.78$ | $230.50 \pm 17.23$ | $161.12 \pm 20.86$ | $-127.92 \pm 4.82$ | $\mathbf{-135.94} \pm 1.71$ |
| $\alpha$KL, $\alpha$=0.0 | learned | $369.30 \pm 0.90$ | $\mathbf{283.92} \pm 0.78$ | $185.28 \pm 0.92$ | $40.38 \pm 3.08$ | $-140.76 \pm 0.42$ |
| $\alpha$KL, $\alpha$=0.0 | log Z | $369.57 \pm 0.60$ | $283.80 \pm 1.15$ | $185.70 \pm 1.16$ | $40.05 \pm 3.00$ | $-138.13 \pm 0.22$ |
| $\alpha$KL, $\alpha$=0.0 | opt. scalar | $369.16 \pm 0.92$ | $283.84 \pm 0.79$ | $186.14 \pm 1.48$ | $39.48 \pm 2.87$ | $-138.16 \pm 0.18$ |
| $\alpha$KL, $\alpha$=0.0 | opt. vector | $\mathbf{369.73} \pm 0.78$ | $283.88 \pm 1.02$ | $\mathbf{186.30} \pm 1.66$ | $\mathbf{41.50} \pm 3.73$ | $-138.04 \pm 0.27$ |

| method | control variate | $\beta = 0.2$ | $\beta = 0.4$ | $\beta = 0.6$ | $\beta = 0.8$ | $\beta = 1.0$ |
|---|---|---|---|---|---|---|
| $\alpha$KL, $\alpha$=0.0 | none | $\mathbf{-135.94} \pm 2.84$ | $-126.93 \pm 4.39$ | $143.81 \pm 22.36$ | $235.06 \pm 17.64$ | $274.11 \pm 31.61$ |
| $\alpha$KL, $\alpha$=0.0 | learned | $-140.69 \pm 0.40$ | $\mathbf{87.20} \pm 21.32$ | $214.36 \pm 44.48$ | $326.74 \pm 34.11$ | $413.30 \pm 35.51$ |
| $\alpha$KL, $\alpha$=0.0 | log Z | $-138.27 \pm 0.31$ | $48.88 \pm 35.28$ | $211.71 \pm 37.90$ | $305.98 \pm 44.27$ | $411.81 \pm 36.92$ |
| $\alpha$KL, $\alpha$=0.0 | opt. scalar | $-138.29 \pm 0.30$ | $80.23 \pm 28.96$ | $\mathbf{219.92} \pm 37.03$ | $\mathbf{327.26} \pm 33.52$ | $\mathbf{427.90} \pm 32.53$ |
| $\alpha$KL, $\alpha$=0.0 | opt. vector | $-138.05 \pm 0.36$ | $74.51 \pm 33.47$ | $211.88 \pm 37.44$ | $326.55 \pm 34.62$ | $405.99 \pm 34.91$ |

Table 3: Expected log-weights of $\alpha$KL with different control variates for ten $15 \times 15$ Ising models with different interaction strengths $\beta$. Note that $\alpha$KL with a learned control variate is equivalent to $\alpha$TB for $\alpha = 0$.

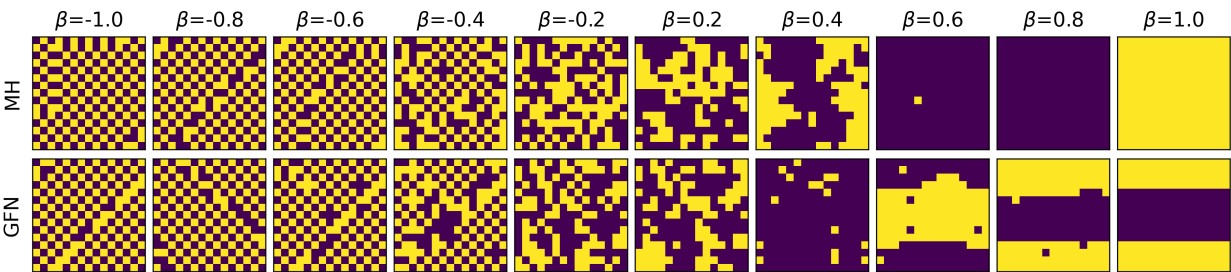

Figure 4: Approximate samples from a 15×15 Ising model generated using MH chains (top row) and the forward model of a trained GFNs (bottom row).

## 5 Conclusion

In this paper, we draw connections between the recent literature on generative flow networks and the literature on variational inference methods. We observe that GFNs can be trained using variational objectives that minimize a divergence between a forward and a backward distribution over trajectories. When minimizing the reverse Kullback-Leibler divergence, the objective is analogous to that used in standard variational inference methods that maximize a lower bound on the log-marginal likelihood (Blei et al., 2017). When minimizing the forward Kullback-Leibler divergence, we obtain a variant of the objective that is commonly used in wake-sleep methods and related approaches (Hinton et al., 1995; Bornschein and Bengio, 2015; Naesseth et al., 2020). It is also possible to optimize a convex combination of the two. These objectives are closely related to the trajectory-balance objective that is typically used when training GFNs. Specifically, the gradient of the RKL is proportional to computing the expected gradient of the TB objective with respect to trajectories that are sampled from the forward distribution. Evaluations on synthetic densities and an Ising model demonstrate that variational objectives for GFNs achieve a comparable performance in terms of the expected log weight relative to variants of the trajectory balance objective. This observation opens up opportunities to explore new variational objectives for GFNs that incorporate credit assignment methods Schulman et al. (2015) as well as importance sampling methods for GFNs based on e.g. variational sequential Monte Carlo (Naesseth et al., 2018) or nested variational inference (Zimmermann et al., 2021).

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

| layer number | layer type | activation function | input shape | output shape | number of parameters |
|:---:|:---:|:---:|:---:|:---:|:---:|
| 1 | Dense | Swish | $B \times D$ | $B \times 256$ | $d \cdot 256$ |
| 2 | Dense | Swish | $B \times 256$ | $B \times 256$ | 65536 |
| 3 | Dense | Swish | $B \times 256$ | $B \times 256$ | 65536 |
| 4 | Dense | None | $B \times 256$ | $B \times 2D$ | $d \cdot 512$ |

Table 4: Neural network architecture of forward model which outputs $2D$ logits, where $B$ is the batch size and $D$ is the size of the state space.

# A    Implementation details

The forward model first selects one of the $\emptyset$-bit positions in the input vector uniformly at random. The bit position is then used to select the 2 corresponding logit values from the output of the neural network (architecture described in 4), which are consecutively used to construct a Bernoulli distribution to draw the new bit value for the selected bit position. In all our experiments we use a fixed backward model, which uniformly at random selects a bit position in the vector representation of the state and deletes it, i.e. sets it to $\emptyset$. Hence, the overall number of parameters is the number of parameters of the forward-model ($N_{\mathrm{params}} = 768D + 131072$) and one additional parameter if the control variate is learned. For all experiments we trained the forward model using the Adam optimizer with a learning rate $\alpha = 1e^{-3}$ and batch size $B = 256$.

