# OpenReview forum: "A Variational Perspective on Generative Flow Networks"
_TMLR — Accepted by TMLR_

### Review · Reviewer_5oNV · 2022-12-11

**Summary Of Contributions:**

The authors study the recently introduced Generative Flow Networks (GFN) model by (Bengio et. al., 2021). Specifically, the authors focus on the training of such networks with the TB objective (Malkin et. al., 2022), equation 1 in this paper. This objective optimizes two models simultaneously: forward and backward distribution models. The training of GFN requires sampling the trajectories from some proposal distribution over trajectories q, but in practice it seems like everyone samples trajectories from the forward model, backward model or their mixtures. THe authors show that in certain cases such training can be viewed as the “mixed” variational inference, i.e., optimizing the weighted sum of forward and reverse KL divergences between the forward and backward model. This point of view opens possibilities to employ the variance reduction techniques from the variational inference. The authors run several experiments to analyse their findings and the effect of the mixing weight.

**Audience:**

Yes

**Broader Impact Concerns:**

-

**Claims And Evidence:**

Yes

**Requested Changes:**

- A more deeper discussion of GFNs with toy examples and use-cases.
- I recommend the authors to add a table summarizing the gradient formulas so that it would be easier for the reader to compare.
- Add more implementation details, at least to appendix.

Some questions:
- The authors consider the case when P,Q have the same parameters. How does this work in practice? More precisely, this boils down to the general question of parameterization & practical implementation.
- I do not understand how large are the final models.
- Do we build the trajectory flow (dyrect acyclic graph) on our own for  for a particular modeling task, or it is already given?
- Why do we need both forward and backward models in GFNs? Are there approaches to optimize only one of them?
- What is the interpretation of the TB objective? Do we want to match the forward model’s and reverse model’s trajectory probabilities?


**Strengths And Weaknesses:**

I would like to emphasize the I have not worked a lot with the graphical models before and, in particular, have not published papers in this area.  I am not very familiar with recent discrete models in his field. Therefore, I am not very familiar with the related work and can not provide a confident assessment of the significance of the current work. In my review, I will try to focus on readability and presentation of the results and hope that the other reviewers will assess the significance and novelty.

- The presentation in the paper assumes that the reader is already familiar with the GFNs and only briefly recalls their main concepts. In general, I think this not true, especially because the GFN model (if I correctly understand) has been introduced only about a year ago (2021). The authors provided all the necessary mathematical details of the in Section 2, which is good. However, I would expect the authors to provide a more detailed introduction of GFNs, possibly with some introductory toy example of a GFN model/visual material, e.g., particular cases of direct acyclic graphs for certain problems. This would notably simplify the understanding of the model to the reader.
- It is not very clear which parts are the authors contribution and are novel and which parts contain known content. For example, I assume that section 3.1 is well known, although it is a little bit unclear from the current text.
- I did not find some implementation details in the paper (and I do not see “supplementary material” on openreview as well), e.g., learning rates, etc. How much computational time do we need to reproduce the experiment? This is not good for reproducibility.
- In Section 4, the authors study the TB objective from the variational perspective. They derive a lot of formulas and, if I understand correctly, show that they is some cases match to the gradients of the losses (2), (3). However, the referencing in this paper is extremely inconvenient. For example , it would be nice if the authors in phrases such as “proportional to the [...]” would also add reference to some line from Section 3. Otherwise, it is hard to compare those sections.
- While the paper focuses on the GFN model, I wonder what are the other models in the field for the tasks in view. The discussion of this aspect is limited. I think it should be added. Otherwise, a reasonable question arises: why do we need to further develop GFNs? I think the authors should provide a clear list of advantages w.r.t. the other (non-GFN) models. The same question is about the experiments: how do GFNs compare with the other models in the field (which can be used as the baselines)? As far as I understand, the experiments mostly focus on testing GFN with various parameter alpha.

---

> ### Author Response · Authors · 2022-12-20
> **Response to Reviewer 5oNV - Part 1**
>
> Thank you for your review! As per the guidelines for TMLR, we will wait for other reviews to come in before uploading an updated manuscript, but we would like to go ahead and discuss how we will incorporate your comments to improve the readability and presentation of the manuscript.
>
> **Why would we need to develop GFNs?** The reviewer is right to point out that this paper does not make a case for GFNs per se, which is not our intent or ambition. We see GFNs as an interesting new model class that is worthy of study. In this context, we would like to understand the connections objectives for training GFNs and variational inference.
>
> **Discussion of Alternative Methods.** There indeed exist other methods that can be used for this task including MCMC variants, Sequential Monte Carlo methods, and methods from variational inference (VI) as well reinforcement learning. Previous work [2,3,4] on GFNs has compared to these methods, and we do not think an experimental comparison will yield substantial new information per se, since the difference relative to previous work lies in the objective, not in the models that are trained with the objective. However, we will incorporate discussion of alternative methods into the background section of the paper.
>
> **Discussion of Background on Generative Flow Networks.** We agree our discussion of background is on the terse side, and we will revise the exposition to make it clear what properties of GFNs are the most relevant to this paper and what aspects of their foundations are described in more detailed introductions elsewhere [1, 2]. While there is a lot to be said about GFNs, their most salient property in the context of this paper is that they define a transition model as a conditional distribution parameterized by an MLP. Focusing on this view allows us to streamline exposition, but we agree that this might be confusing. We will therefore revise section 2 to provide more context. We will also add a figure which showcases how the directed acyclic graph (DAG) representation is related to the generative model and flow formalism.
>
> **Definition of the DAG.** While we explicitly define the DAG structure in our experiment section, this is generally not needed. The graph structure is implicitly defined by defining the forward and backward transition models, similarly to a Markov Decision Process (which can be also thought of in terms of a graph) which is defined in terms of the state and action spaces, the reward, and the transition model. We will clarify this in the manuscript.
>
> **Differentiating our contribution from background material.** As the reviewer notes, the material presented in section 3 is indeed well known, and we consider it background. Our contributions lie in section 4 and the experiments, which identify equivalences between the described variational objectives and the trajectory balance objective and in the consequent realization that commonly used variance reduction techniques from variational inference can also be applied to the GFN settings. To make this more explicit, we will restructure the sectioning by introducing a Section 2 on Background, which will contain subsections 2.1 (GFNs; the old section 2) and 2.2 (VI; the old section 3).
>
> **Implementation details.** Thank you for pointing out these missing details. We will upload a supplementary material document with the implementation details and used hyperparameters, including learning rates, architectures, and number of parameters. In our experiments we use a fixed backward model, hence the overall number of parameters is the number of parameters of the forward-model (plus one additional parameter if the control variate is learned). For both experiments the GFN is parameterized by an MLP with layers [d x 256 x 256 x 256 x (3*d)], where d is the dimensionality of the state space.
> - Experiment 1: d=16; 147,456 parameters
> - Experiment 2: d=15**2; 361,472 parameters
>
> **Summary Table and missing references to equations.** We think that adding a table which summarizes the gradients formulas is an excellent idea. We will add this to section 4 in the main text. We will also add equation references to the corresponding formulations in the text.
>
> Part 1/2

---

### Review · Reviewer_LtBg · 2023-01-11

**Summary Of Contributions:**

 In this work the authors introduce a variational objective for training GFNs, which is a convex combination of the reverse- and forward KL divergences, and compare it to the trajectory balance objective when sampling from the forward- and backward model, respectively.
 In certain settings, variational inference for GFNs is equivalent to minimizing the trajectory balance objective, in the sense that both methods compute the same score-function gradient. The proposed variational objective is evaluated numerically  on two synthetic  data.

**Audience:**

Yes

**Claims And Evidence:**

Yes

**Requested Changes:**

The authors should either justifying  the method on    real  data set or giving  theoritical study.

**Strengths And Weaknesses:**

Stengths: The idea is ok.
Weakness: The proposal in this paper need more justification on  real  data set or in theory.

---

### Review · Reviewer_QqZW · 2023-02-02

**Summary Of Contributions:**

This paper highlights links between two frameworks, the recent GFlowNet framework and variational methods. In particular it shows that under certain conditions (notably the choice of objective and data distribution) the two frameworks induce the same gradients, namely the on-policy GFlowNet Trajectory Balance objective corresponds to the reverse KL divergence objective when $P_B R/Z$ is taken as the prior and $P_F$ is taken as the posterior.

The paper then suggests that the variance reduction techniques used in variational methods can therefore be translated to GFNs. This is shown both in theory and in a series of experiments that suggest control variates may have an impact on training GFNs.

**Audience:**

Yes

**Broader Impact Concerns:**

No concerns.

**Claims And Evidence:**

Yes

**Requested Changes:**

In section 4 the authors suggest that the objective of Zhang et al., $L_{\alpha TB}$ corresponds to sampling from either $P$ or $Q$ with probability $\alpha$. This doesn't seem correct, at least, as far as I can tell this is not what Zhang et al. propose, since the backward trajectories that they use are obtained by first sampling $x$ _uniformly_ from a dataset, and only then sample a backward trajectory from $P_B$. This is because they assume an identical reward/energy for all instances in the data (which they are jointly learning), but it is not exactly sampling from $P_B$. In fact, in the general case for GFNs, as far as I know there is no tractable way to sample from $P_B$.

Note that this remark doesn't invalidate the theoretical results shown here, the equivalences drawn still hold -- but this makes some of the empirical work a bit incomplete to me. If we assume that $\pi_T$ is given, then there's not really any point in training a model that essentially recovers $\pi_T$. What I understand to be one of the strengths of GFN is that it is off-policy; it might have been interesting to compare using different distributions.

The Figure 1/Table 1 results are suspicious, not in bad sense, but in that something feels like it should be investigated more. The gap between KL and TB seems to be problem dependent, it only appears for the two spirals. This might be saying something about the optimization dynamics (as the authors point out) or the combo of dynamics + function approximation, which it would be nice to investigate.

Finally, I understand that this is concurrent work with Malkin et al., but it might be worth demarcating the differences more clearly. Also note that the $\alpha KL$ objective is not exactly novel, although its treatment here matters, see _On a generalization of the Jensen-Shannon divergence_ by Frank Nielsen.


**Strengths And Weaknesses:**

The paper is well written and felt easy to understand. The contribution is fairly clear: there are links between GFNs and variational methods. The paper feels a bit weak in that it doesn't go further than that, it establishes equivalences and differences on a theoretical level, but the empirical follow up is minimal. There is no characterization of the settings where these differences matter, or of where trade offs between methods appear.

---

> ### Author Response · Authors · 2023-02-09
> **Response to Reviewer QqZW**
>
> Thank you for your review!
>
> **Clarification regarding sampling of backward trajectories.** We agree that in general there is no tractable way to sample directly from $P_B(x \mid s_f) = R(x) / Z$ (we use $s_T = x$ here). While we do have access to online samples from the target distribution in our first experiment (following Zhang et al.), it is generally sufficient to have access to a large enough data set $X=\\{x_i\\}^n_{i=1}$ with $x_i \sim P_B(\cdot \mid s_f)$. In this case, we can uniformly subsample the data, in the same way as described by Zhang et. al and use each data point as a starting point to sample a full backward trajectory according to $\prod_{t=0}^{T-1} P_B(s_t \mid s_{t+1})$. The density of the complete trajectory $P(\tau) = R(x) / Z\prod_{t=0}^{T-1} P_B(s_t \mid s_{t+1})$ is computed as a function of the reward $R(x)$ of each individual data point, hence there is no assumption of "identical reward/energy for all instances in the data". We will clarify how the data and complete backwards trajectories are sampled in the experiment section.
>
> **Significance and extent of empirical findings.** While we were hoping to find more substantial overall performance differences, it is not the case that there are no differences in all settings. For $\alpha=1$, training with the variational objective, which reduces to the forward KL-divergence in this setting, significantly outperforms training with the trajectory balance objective (see Figure 1 and Table 2). However, the best performance is generally seen for $\alpha < 1$, and the superior performance of the forward KL does not carry over to the cases where a mixture of samples from the forward and backward model are used, even for $\alpha = 0.75$ where 75% of samples are obtained from the backward model. Despite extensive experimentation with the synthetic density model (D=32) and the more challenging Ising model (D=225), we were not able to find settings (other than for $\alpha=1$ as described above) where the variational objective or trajectory balance objective consistently outperform the other by a significant margin. For $0<\alpha<1$, the $\alpha$TB objective seems to perform slightly better on average, which is consistent with the findings of concurrent work by Malkin et al.
> Training _bigger_ models makes it increasingly more difficult to statistically evaluate the effect of different combinations of variables. For the presented numeric evaluation of the Ising model we trained 10 independent models for 50 different combinations of inverse temperatures $\beta$ and control variates, which equates to 500 models for each, the $\alpha KL$ and $\alpha TB$ objective. Finally, we consider the main contribution of the paper to be establishing the connection between GFNs and variational inference and developing theoretical equivalences.
>
> **Related work on variational objectives**
> Our objective is a convex combination of the forward- and reverse KL-divergence, which is different from the (vector-skew) $\alpha$-Jensen-Shannon divergences described in  _On a generalization of the Jensen-Shannon divergence_. We do not claim that our objective is novel, indeed for or $\alpha=0.5$, our objective reduces to Jeffreys divergence. We will make sure to mention this connection in the manuscript when we introduce the variational objective.
>
> **Differences to concurrent Work.**
> We will add clarifying remarks to the discussion on concurrent work in our related work section.

---

> > ### Comment · Reviewer_QqZW · 2023-02-10
> > **Follow up**
> >
> > Thanks for the response. The update to the paper looks nice.
> >
> > > In this case, we can uniformly subsample the data
> >
> > This is what I'm not sure I understand. If you uniformly sample from the dataset (presuming that different $x_i\in X$ have different rewards $R(x_i)$), then you're not sampling from $P(\tau) = R(x) / Z\prod_{t=0}^{T-1} P_B(s_t \mid s_{t+1})$, because sampling from that distribution would imply first sampling $x$ in proportion to $R(x)$, then sampling a trajectory backwards from a distribution over parents. In that sense, if you sample uniformly, with probability $1/n$, it's like assuming that all samples in $X$ have equal reward. Wouldn't sampling from the dataset non-uniformly $\propto R(x)$ "be more like $P_B$"?

---

> > > ### Author Response · Authors · 2023-02-13
> > > **Re: Follow up**
> > >
> > > Thank you for your positive feedback!
> > >
> > > It is assumed that the data $X=\\{ x_i \\}_{i=1}^N$ was sampled according to the target density $P_B(x \mid s_f)$ under which each data point $x_i$ has probability density $P_B(x_i \mid s_f) = R(x_i)/Z$. Therefore, data points with higher density are represented more often in $X$ than data points with lower density. Technically, $X$ is used to define an empirical density $\\hat P_B(x \mid s_f) = \\frac{1}{N} \\sum^N_\{i=1\}\delta(x - x_i)$, which approximates the target density (for large enough N). Hence, drawing uniform samples from the data set is equivalent to sampling from $\\hat P_B(x \mid s_f)$ and approximates samples from the true target density $P_B(x \mid s_f)$.

---

### Comment · Action_Editors · 2023-02-03
**The discussion phase**

Dear reviewers and authors,

The discussion phase has finally started! Now we have about 2 weeks after which I would like to kindly ask all reviewers to provide their recommendation.

Please read all reviews and authors' rebuttals.

All the best,
Your AE

---

### Author Response · Authors · 2023-02-09
**Updated manuscript**

Dear reviewers,

Thank you again for your reviews! We have uploaded an updated manuscript which incorporates the requested changes. To provide an easy overview we have listed all relevant changes in the revised submission and highlighted the relevant changes in the manuscript in red.

---

### Decision · Action_Editors · 2023-03-13

**Recommendation:** Accept as is

**Comment:**

There is a consensus among two reviewers that the paper is well-written and presents interesting ideas. The third review is too short to take it into account. The summary of the reviews:
- The paper is well-written.
- The links between GFNs and VI are clear.

The authors did a great job in the rebuttal. They followed (most of) the points raised by the reviewers, thus, it seems that the claims are well supported with evidence now.

In my assessment, the paper fulfills all requirements to be accepted as is.


**Audience:**

Generative Flow Networks constitute a new class of models that attract more attention due to their interesting properties. This paper contributes to further understanding of this class of models. Thus, there will be definitely an interest within the TMLR community.

**Claims And Evidence:**

Claims:
1) In certain settings, optimizing the trajectory balance objective is equivalent to optimizing a forward- or reverse Kullback-Leibler divergence.
2) The cases in which the TB objective is equivalent to the corresponding variational objective are identified and this insight is leveraged to employ variance reduction techniques from variational inference.
3) To evaluate the presented theoretical findings and the empirical performance of the trajectory balance and the corresponding variational objective, a set of experiments is carried out.

Evidence:
- Ad 1) and 2): Section 3.
- Ad 3): The experiments comprise of synthetic densities and an Ising model, a widely used model in physics.